# Synthesis and Characterization of Polymer-Based Membranes for Methotrexate Drug Delivery

**DOI:** 10.3390/polym15214325

**Published:** 2023-11-04

**Authors:** Ionela-Amalia Bradu, Titus Vlase, Mădălin Bunoiu, Mădălina Grădinaru, Alexandru Pahomi, Dorothea Bajas, Mihaela Maria Budiul, Gabriela Vlase

**Affiliations:** 1Research Centre for Thermal Analysis in Environmental Problems, West University of Timisoara, Pestalozzi Street 16, 300115 Timisoara, Romania; ionela.bradu@e-uvt.ro (I.-A.B.); titus.vlase@e-uvt.ro (T.V.); alexandru.pahomi@e-uvt.ro (A.P.); dorotheabajas@gmail.com (D.B.); mihaela.budiul@e-uvt.ro (M.M.B.); 2ICAM–Advanced Environmental Research Institute, West University of Timisoara, Oituz Street 4, 300233 Timisoara, Romania; madalin.bunoiu@e-uvt.ro (M.B.); madalina.mateescu@e-uvt.ro (M.G.); 3Faculty of Physics, West University of Timisoara, B-dul V. Parvan No. 4, 300223 Timisoara, Romania

**Keywords:** methotrexate, membranes, drug delivery, polymers, synthesis

## Abstract

Methotrexate or amethopterin or 4-amino-N10-methyl pteroylglutamic acid is used for treating autoimmune diseases, as well as certain malignancies. Drug delivery systems, which are based on biopolymers, can be developed to improve the therapeutic and pharmacological properties of topically administered drugs. Biopolymers improve the therapeutic effect of drugs, mainly by improving their biodistribution and modulating drug release. This study presents the synthesis of membranes based on anionic polysaccharides and cationic polysaccharides for transdermal delivery of the active ingredient methotrexate, as well as a compatibility study between methotrexate and each of the components used in the prepared membranes. The obtained membranes based on different marine polysaccharides, namely κ-carrageenan and chitosan, for the release of the active ingredient methotrexate were characterized using techniques such as TG, FTIR, UV–Vis spectrophotometry, FTIR microscopy, water absorption capacity, water vapor permeability, and biodegradation rate. Following the studies, the membranes suitable for the transdermal release of the active substance were validated.

## 1. Introduction

The field of biopolymers has more than half a century of development and innovation due to their rather complex structure, multiple physiological functions, wide variety, and relatively low cost. Biopolymers can be biosynthesized from living organisms, such as algae, or chemically synthesized from biological materials [1,2].

Drug delivery systems, which are based on biopolymers, can be developed to improve the therapeutic and pharmacological properties of topically administered drugs [3]. Biopolymers have received significant attention in a variety of applications that require biodegradable and sustainable solutions. The development of drug delivery strategies to boost the activity of bioactive substances is still a crucial strategy for achieving disease treatment, and progress in this field has been enormous. Advantages of using biopolymers include: support for the growth of new tissue, inhibition of the activity of cells, guided tissue response, enhancement of the cell–cell interaction and the resulting stimulation of the cells, inhibition of activation or attachment of cells, and inhibiting a biological reaction. Biopolymers improve the therapeutic effect of drugs, mainly by improving their biodistribution and modulating drug release [4,5]. Marine polysaccharides with different biological properties such as alginate, chitosan, or carrageenan allow for the discovery of new pharmaceutical formulations or active ingredients suitable for biomedical applications [6]. 

In another study published by our research group, we have obtained and characterized alginate-based membranes for the transdermal release of methotrexate [7]. Our current study aims to further develop the previous study by investigating the possibility of using membranes based on other marine polysaccharides, such as κ-carrageenan and chitosan, for the transdermal release of methotrexate. Since alginate is an anionic polysaccharide [8], the present study aimed to characterize and compare pharmaceutical formulations for the release of methotrexate based on another anionic polysaccharide, namely κ-carrageenan [9], or a cationic polysaccharide, namely chitosan [10]. Other research groups have published studies on the release of methotrexate, e.g., through hydrogels [11] or magnetic nanoparticles [12]. 

Several polysaccharides are known under the name carrageenan, which are obtained from red seaweeds of the families *Gigartimaceae*, *Solieriaceae*, *Hypneaceae*, and *Furcellariaceea* from the class *Rhodophyceae*. They are hydrophilic colloids that can be classified into six basic forms depending on their sulfate content, origin, and solubility: kappa (κ-), iota (ɩ-), lambda (λ-), mu (μ-), nu (ν-), beta (β-), and theta (θ-) carrageenan. From a chemical point of view, carrageenans have a linear polygalactopyranose structure in which part of the hydroxyl groups are esterified with sulfuric acid [13,14,15]. k-carrageenan (**Car**) is a linear sulfated polysaccharide with repeating D-galactose and 3, 6-anhydro-D-galactose units. The anionic sulfate pendant group may interact with cationic drugs between electrostatic interactions [6].

Chitosan (**Chi**) is a natural polysaccharide obtained from the deacetylation of chitin, present in the exoskeletons of marine crustaceans, including shrimp and crab. It is the second-most abundant natural polysaccharide next to cellulose. Like carrageenan, it has many advantages over other polymers, such as nontoxicity, biocompatibility, and biodegradability. It has a primary, cationic amino group responsible for many of its unique properties, such as pH sensitivity, in situ gelation, mucoadhesion, and increased permeation. Active amino groups in chitosan are responsible for its versatility and cationic character; it acts as a reactive site for the attachment of different groups with mild reaction conditions [16,17,18]. In the development of a controlled-release drug delivery system, it has been extensively studied due to the fact that it facilitates the transmucosal absorbtion of drugs because the negatively charged mucosal surface interacts with the positive electrostatic charges of chitosan [3]. 

Methotrexate (**MTX**) (see Figure 1) or amethopterin or 4-amino-N10-methyl pteroylglutamic acid [19] is used for autoimmune diseases such as psoriasis and rheumatoid arthritis, as well as certain malignancies. Oral use of methotrexate causes gastrointestinal and hepatic toxicities. Oral administration of methotrexate leads to severe side effects such as abdominal pain, anemia, nausea, thrombocytopenia, or depressive disorders. Considering all these disadvantages, it is desirable to administer methotrexate topically [20].

Transdermal drug delivery offers advantages such as avoiding drug degradation in the gastrointestinal tract (GIT), maintaining the drug at the therapeutic concentration for a longer period of time, minimizing dosing frequency, better tolerability, and improved patient compliance [21].

In our study, membranes based on anionic polysaccharides and cationic polysaccharides for transdermal delivery of the active ingredient **MTX** were developed and characterized, employing current polymers used in drug delivery systems: polyvinyl alcohol **PVA** or polyvinylpyrrolidone **PVP**. **PVA** and **PVP** were added to the composition of the membranes in order to increase the hydrophilic properties of the membranes. In addition to the polymer, glycerol and Span80 as plasticizers were also used in all pharmaceutical formulations. 

The prepared membranes, based on different marine polysaccharides, were characterized using techniques such as TG, FTIR spectroscopy, UV–Vis spectroscopy, and different characterization tests.

## 2. Materials and Methods


**Chemicals and Reagents**


Active substance used in this study was Methotrexate **MTX**, 10 mg/mL, produced by Calbiochem (Lot: D00135520 MTX-EMD Chemicals Inc., San Diego, CA, USA). Biopolymers used for membrane formulation were chitosan (**Chi)** from Thermo Scientific (Waltham, MA, USA), M.W. 100,000–300,000, CAS: 9012-76-4, Lot: A0437756; and k-carrageenan (**Car**) from Acros Organics (Geel, Belgium, CAS:11114-20-8). The plasticizers were represented by glycerin (**G**) sold by CHIMREACTIV (Ion Creanga, Romania) and Span 80 (**S**) from Sigma-Aldrich, St. Louis, MO, USA, PC-1002614428 S6760-250 mL, Lot #MKCF4338. Polyvinylpyrrolidone (**PVP**) M.W. 4000 powder sold by CALBIOCHEM, Merck, Darmstadt, Germany (Lot: BBCV7638, CAS: 9003-39-8) and polyvynil alcohol (**PVA**) from Merck, Darmstadt, Germany, S7316066 641 were used also as polymers. Lysozyme from chicken egg white (100,000 U/mg) was purchased from Sigma-Aldrich, Lot BCBP7829V, Darmstadt, Germany.


**Synthesis of Membranes**


The first stage in the synthesis of the membranes was represented by obtaining the stock solutions used in their formulation. Thus, to obtain the biopolymer matrix, a 1% k-carrageenan solution was used, obtained by dissolving it in distilled water, and 1% chitosan was obtained by dissolving it in distilled water and 5% acetic acid. For **PVA** and **PVP** solutions, 500 mg of polymer was dissolved in 20 mL of distilled water under continuous magnetic stirring for 2 h.

For the preparation of membranes containing Glycerol (**G**) as a plasticizer, the procedure was as follows: **G** was mixed with k-carrageenan and chitosan, respectively, from the stock solution, then the **PVA/PVP** polymers and **MTX** were added and stirred using a magnetic stirrer and poured into sterile petri plates and left to dry in fresh air before analyses. A similar procedure was used for membranes containing Span 80 as a plasticizer. Table 1 summarises the synthesis of membranes.


**FT–IR Spectrum**


Infrared spectra were collected using a Shimadzu IRTracer-ATR in the range of 400–4000 cm^−1^ at 4 cm^−1^ optical resolution and 20 scan repetitions per analysis. ATR accessory equipped with a single reflection diamond ATR crystal on ZnSe plate was used for all of the analysis. 

**Thermogravimetric analysis (TGA**)

Thermogravimetric analysis (TGA) was carried out from 25 to 400 °C under synthetic air atmosphere (20 mL/min) in a TG/DTG Mettler TOLEDO model TGA/DSC3^+^. Samples were analyzed in aluminum pan at 10 °C/min as heating rate. For calibration on TGA/DSC3^+^, calibration standards of In (12.29 mg), Al (5.52 mg), Au (41.01), and Pd (26.67 mg) were used.


**UV–VIS Spectrophotometry**


Samples were analyzed using an T90 + UV–Vis Spectrophotometer with a double beam, in the range of 200–400 nm, with a scan rate of 100 nm min^−1^, and UV–VIS data interval of 1.00 nm.


**Membranes characterization**


To characterize the membranes physico-chemically, tests of the following type were used: swelling ratio, water vapor permeability, biodegradation rate.

Swelling ratio (*SR* (%))

The swelling property of the membranes was measured by cutting dried samples into pieces of 10 mm × 10 mm and immersing them into phosphate buffered saline PBS (pH= 7.4) at 37 °C. At measurement point (24 h), the samples were removed from the solution, blotted with filter paper to remove adsorbed water on the surface, and weighed.

The swelling ratio of water absorbed was calculated from the formula:$$SR(\%)=\frac{W_{s}-W_{d}}{W_{d}}\times 100$$
where *W_d_* is the weight of the dried membrane; and *W_s_* is the weight of swollen membrane [22].

Water vapor permeability

Evaporation of water through the test membranes was monitored by measurement of water loss from the system at 37 °C in 24 h. The membranes were fixed with a rubber band on the opening of a testing tube filled with distilled water. The water vapor transmission rate (*WVTR*) was determined by the following formula:$$WVTR\left(g\times m^{-2}\times d^{-1}\right)=\frac{W_{T}-W_{0}}{t\times A}$$
where *W*_0_ is the weight of the initial system; *W_t_* is the weight of the system at time *t*; *t* is the measuring time (1 day); *A* is the area of the opening of the test tube [23].

Biodegradation rate

The biodegradation rate (*MR* (%)) of the membranes were studied by monitoring the weight loss of the samples in a time period. Wet samples of each membrane of 10 mm × 10 mm with known weight (*W*_0_) were immersed in a 3 mg/mL lysozyme/PBS solution and kept in an incubator at 37 °C. At predetermined time points, samples were taken out and weighed (*W_t_*) after blotting off the residual incubation medium. The biodegradation kinetics were considered as mass loss in the period of observation [22].
$$MR\left(\%\right)=\frac{W_{t}}{W_{0}}\times 100$$

## 3. Results and Discussion

### 3.1. Membrane Formulation

In accordance with the general recipe outlined above, membranes with and without the active substance were created in order to compare the outcomes of the techniques used to create the two types of membranes. This comparison highlighted the presence of **MTX** in the membranes. At the same time, it was intended to compare membranes based on **Chi** or **Car** with another biopolymer, to see which membranes met the criteria for elasticity, homogeneity, bioavailability, and lack of interactions between polymer components and the active ingredient.

Following the syntheses, it was found that the membranes containing **Span 80** as the plasticizer are not elastic and some membranes are inhomogeneous; therefore, they were not analyzed using physicochemical methods. The appearance of the membranes obtained is presented in Table 2.

### 3.2. FTIR-ATR Analysis

As mentioned above, membranes based on the two marine polysaccharides, namely κ-carrageenan and chitosan, were prepared. The FTIR spectrum of **MTX** is shown in Figure 2 and displays characteristic bands according to the literature [24,25]: the O-H stretching vibrations of crystallized water at 3351 cm^−1^, the N-H stretching vibration at 3190 cm^−1^, and the C-H vibrations of the CH_3_ group at 2950 cm^−1^. The next group, from 1670 to 1601 cm^−1^ is assigned to the C=O stretching vibrations (C=O stretching of carboxylic group and –C=O stretching of amidic group), respectively. The bands from 1555 to 1508 cm^−1^ are associated with the N-H of amide group. The frequency of 1097 cm^−1^ is attributed to C-N stretching [26]. 

Table 3a shows the wavenumber of values for the main peaks in the FT-IR spectra of the membrane based on κ-carrageenan with the active ingredient **MTX**. 

The κ-carrageenan-based membranes also contain glycerol, **PVA**, or **PVP** and the active ingredient **MTX**, so the characteristic peaks of all compounds can be found in Figure 3. The compatibility study was also performed for the matrices with polyssacharide and glycerol and for the matrices with polymer or **MTX**. 

The main characteristic peaks of these two components could be found in all spectra. The specific peaks of this polysaccharide were as follows: 3316 cm^−1^ for the vibrations of the hydroxyl groups, while the vibrations of the C-H bonds in the saccharide units are found at 2940 cm^−1^. The frequency of 1592 cm^−1^ is characteristic of the aldehyde carbonyl that was present in the compound. The stretching vibration of the S=O bond showed an absorption maximum at 1202 cm^−1^. The peaks found at 1032 cm^−1^ and 919 cm^−1^ are specific for the C-O stretching vibration, both from the C-OH and C-O glycosidic bond (linkage of 3,6-anhydro-D-galactose units) [27]. The κ-carrageenan is a sulfated galactan, it shows a “fingerprint” at 827 cm^−1^ due to νC-OSO_3_H, and the lack of this peak could indicate a chemical modification of k-carrageenan. The peak found at 920 cm^−1^ is attributed to the vibrations of the C-C skeleton in the case of glycerol. 

From the data presented in Table 3a, it can be seen that the κ-carrageenan-based membrane with **PVA** and **MTX** contains all the characteristic peaks of the active ingredient. There is no interaction in this type of membrane.

In the membrane containing PVP and **MTX**, it is observed that the N-H stretching vibrations overlap with the O-H stretching vibrations. The characteristic bands of the active ingredient **MTX** are present, indicating the existence of C=O, N-H, and C-O within the membrane. 

For the chitosan-based membrane, the FTIR spectrum is shown in Figure 4. This type of membrane contains also glycerol, **PVA,** or **PVP,** and the active ingredient **MTX**. As in the previous type of membrane, the compatibility study was also performed for the matrices with polysaccharide and glycerol and for the matrices with polymer or **MTX**. Table 3b shows the wavenumber of values the main peaks in the FTIR spectra of the membrane based on chitosan with the active ingredient **MTX**. 

The main absorption peaks of chitosan are at 1650 cm^−1^ assigned to the C=O stretching of amide I, the N-H bending of amide II at 1557 cm^−1^, and the C-N stretching of amide III at 1323 cm^−1^. The peak at 3230 cm^−1^ is due to amine N-H symmetric vibration. The typical C-H vibration is at 2925 cm^−1^ [28,29]. Also, for this type of membrane, the peak found at 920 cm^−1^ is attributed to the vibrations of the C-C skeleton in the case of glycerol.

The FTIR spectrum of the chitosan-based membrane with PVA and **MTX** does not show the characteristic peaks of the C=O stretching vibration specific for the active ingredient. The peak at 1734 cm^−1^ assigned to the C=O stretching vibration is due to the stretching of the acetate group of **MTX**. Also, the peak associated with the C-N stretching vibration of **MTX** is present at 1100 cm^−1^ [30]. 

In the case of the FTIR spectrum of the chitosan-based membrane with **PVP** and **MTX,** it is also observed that the N-H stretching vibrations overlap with the O-H stretching vibrations. The O-H stretching vibration is found at 3300 cm^−1^. The C=O stretching vibration is also found at 1652 cm^−1^. The band corresponding to the N-H bending of an amidic group is present at 1558 cm^−1^. The band corresponding to the C-O stretching from a carboxylic group is found at 1459 cm^−1^. In this case, the peak of the C-N stretching vibration is present at 1100 cm^−1^.

There were no significant interactions found in the membranes’ FTIR analysis. The only interactions present in the membranes **CarGPVPMTX**, **CarGPVAMTX**, **ChiGPVPMTX**, **ChiGPVAMTX** are those of the ionic type between the −OH and −NH_2_ groups, strengthened by the formation of −NH_3_^+^…O^−^ bonds that explain the minor shifts in the wavenumber.

### 3.3. TG/DTG Analysis

Thermogravimetric data obtain for membranes based on polymers are graphically represented in Figure 5 and Figure 6 and are compared with both types of membranes in which **MTX** was incorporated, and with the control membrane without the active substance. The thermal behavior of the samples **MTX (SA)** and **MTX** (obtained by recrystallization from the injectable solution) was reported in our previous study. According to this, it was observed that the thermal degradation occurs in four endothermic processes. The first and second thermal events are accompanied by a loss of mass up to T = 132.5 °C, and a new endothermic peak appears at 196 °C, indicating the melting of the MTX in order of the literature (195 °C) [7].

From the TG and DTG curves for the membrane based on **CarG,** shown in Figure 5 and Table 4a, we can conclude the following:

The **CarG** membrane shows a single thermal process with a significant mass loss of 57.70%. For **CarGPVP** and **CarGPVA**, it can be observed that the first membrane shows a lower mass loss of 48.41%, while the membrane with the other polymer present two thermal events with a total mass loss of 63.51%. 

The last three membranes, namely **CarGMTX**, **CarGPVPMTX,** and **CarGPVAMTX** showed a single process. It can be observed that in the case of the **CarGPVAMTX,** decomposition starts at lower temperature, namely 197 °C, compared to the membranes containing **PVP** as polymer. Of all the membranes, the **CarGPVPMTX** membrane exhibited the lowest percentage mass loss at 37.12%. 

From the TG and DTG curves for membranes based on **ChiG**, shown in Figure 6 and Table 4b, we can say the following:

The **ChiG** membrane shows two thermal decomposition steps with a total mass loss of 66.57%. For **ChiGPVP** and **ChiGPVA**, it can be observed that in both cases the decomposition occurs in two steps and the total mass loss is the highest for the **ChiGPVA** membrane with 62.83%. 

The decomposition process of the membrane without the polymers **ChiGMTX** begins at a higher temperature of 229 °C and in a single step, compared to the membranes with **PVA** and **PVP.** Of all the membranes, the **ChiGPVPMTX** membrane exhibited the lowest percentage mass loss of 51.09%. The results of the thermal analysis support the results obtained from the FTIR analysis.

### 3.4. UV–VIS Results

The UV−VIS evaluation was intended only as a qualitative assessment of **MTX** availability in the composition of the designed membranes. All measurements were performed in a 10 mm UV/VIS spectroscopy cell at room temperature using 0.9% NaCl with a pH of 6.7–7.0 as a blank. The pH of the 0.9% NaCl solution was corrected with 0.1 M NaOH. The concentration of **MTX** standard solution was 1 mg/mL. Only four prepared membranes were evaluated using UV−VIS analysis since they were validated by the other techniques, which also met the characteristics related to appearance and homogeneity. The presence of the **MTX** was observed at around 270–280 nm, considering the available peak from Figure 7 and Figure 8. 

### 3.5. Images Obtained with AIM-9000 Shimadzu Microscope in Visible Spectrum

The images obtained with the help of the microscope (Figure 9) indicate a homogeneous distribution of both the polymer structure in all four cases of the membranes that were validated using the physico-chemical study within the article. 

### 3.6. Membrane Characterization


*Swelling ratio (SR (%))*


The ability of polymeric systems to absorb the solution of interest, measured in mass or volume, is known as the swelling degree. The swelling of polymers is the mass transfer phenomenon of fluid diffusion into the material, which facilitates the transfer of the active substance from the polymer matrix. Swelling behavior of the samples was investigated in buffer solution, pH 7.4 at 37 °C. The results obtained are presented in Table 5. 

In case of membranes based on k-carrageenan, their dissolution in the used buffer was observed, so SR (%) could not be calculated. For membranes based on chitosan, the maximum swelling rate was obtained for **ChiGPVAMTX.** Biopolymer membranes have an alteration in swelling capacity due to the addition of compounds such as drugs, active compounds, and plasticizers. Although chitosan contains hydrophilic groups (-NH_2_ and -OH) that can increase the affinity for water molecules through the formation of hydrogen bonds, the addition of glycerol in the membrane structure increases the distance between chitosan chains, increasing the number of free hydrophilic groups, which should lead to improved absorption properties of the water. The difference between the SR (%) for the **ChiGPVAMTX** and **ChiGPVPMTX** membranes can be argued with the presence of hydrophilic groups in the **PVA** structure, groups that are not present in the **PVP** structure.


*Water vapor permeability*


Water vapor transmission rate (*WVTR*) was determined for membranes **CarGPVAMTX**, **CarGPVPMTX**, **ChiGPVAMTX**, and **ChiGPVPMTX** in order to study the vapor permeability. The permeation of water vapor through the membranes involves the adsorption and diffusion processes. Table 6 summarises the WVTR obtained. 

In the study conducted by Lanke et al. [31], the evaporative water loss for normal skin, burns, and granulating wounds at a surface temperature of 35 °C was 204 g m^−2^ per day for normal skin, and up to 5138 g m^−2^ for seriously injured skin. The results obtained are in the middle of the range. 


*Biodegradation rate*


The biodegradation studies of the membranes were conducted by monitoring the weight loss of samples. The lysozymic degradation profiles of the chitosan and carrageenan membranes are shown in Figure 10. The study on the membranes was carried out in vitro by incubating the samples in PBS (pH 7.4) containing 3 mg/mL of lysozyme solution at 37 °C.

In the case of membranes containing **PVA**, a degradation profile is observed that takes place in two phases, phase I (fast degradation) and phase II (slow degradation). The carrageenan–**PVA** membrane degraded approximately 75% after 1 day of lysozymic degradation and 85% after a period of 5 days. For the chitosan–**PVA** membrane, a degradation of 46% is observed in the first 24 h, and after 5 days the degradation is 71%. 

For the carrageenan–**PVP** membrane, rapid degradation is observed in the 24–48 h interval, with a degradation of 58%, and after 5 days of enzymatic exposure, the degradation becomes constant. The degradation of the chitosan–**PVP** membrane is a constant process, the mass loss at the end of the period under study being 56%.

The difference between the degradation profiles of the two polymers, chitosan and carrageenan, can be explained based on their solubility, more precisely the dependence of chitosan on an acidic pH. Also, the hydrophilic groups of **PVA** influence the degradation behavior of the membranes. The results obtained from the swelling ratio, water vapor permeability, and biodegradation tests suggest that chitosan-based membranes are a suitable choice for MTX-containing formulations, especially those based on chitosan–**PVA**.

## 4. Conclusions

This study presents the synthesis and characterization of several types of chitosan-based and carrageenan-based membranes to choose the best membrane that can be used for transdermal delivery of MTX. Along with biopolymers, **PVA** and **PVP** were also added with the aim of increasing the hydrophilic properties of the membranes. In the case of this study, the synthesis and analysis of a consistent number of membranes and the validation based on an experimental protocol of the best formulations, which avoid the interactions of the active substance with the components of the membranes, were achieved.

This study was then extended to the obtained membranes and compared the membranes without active substances. In order to argue the lack of interactions between the base components in the membranes and the base substance, studies by TG/DTG analysis and FTIR methods were performed, which concluded that some membrane bases are unsuitable for MTX incorporation. The results obtained for **CarGPVAMTX**, **CarGPVPMTX**, **ChiGPVAMTX,** and **ChiGPVPMTX** membranes revealed the presence of MTX at a similar wavelength already reported in the literature in our previous study. Following the physical–chemical compatibility studies, the membranes **CarGPVAMTX**, **CarGPVPMTX**, **ChiGPVAMTX,** and **ChiGPVPMTX** were validated.

The four membranes validated by the physicochemical study were subsequently tested to optimize and validate the transdermal release of the active substance. Following swelling studies, water vapor permeability and biodegradation tests suggest that chitosan-based membranes are a suitable choice for MTX-containing formulations, especially chitosan–**PVA**-based ones.

## Figures and Tables

**Figure 1 polymers-15-04325-f001:**
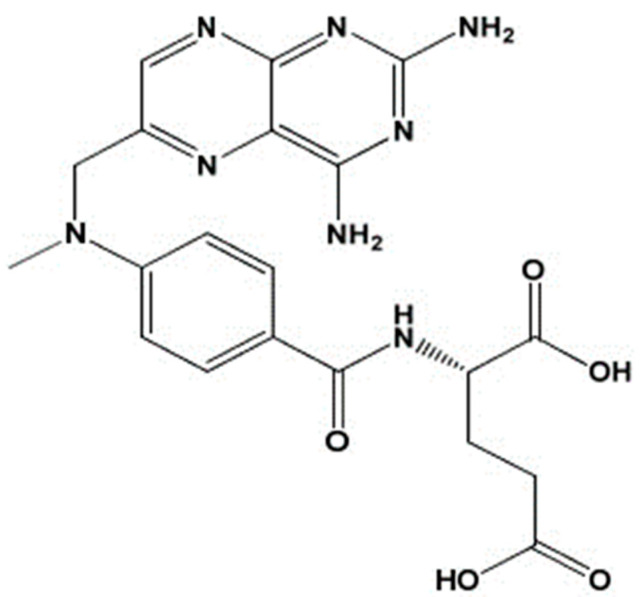
Chemical structure of **MTX**.

**Figure 2 polymers-15-04325-f002:**
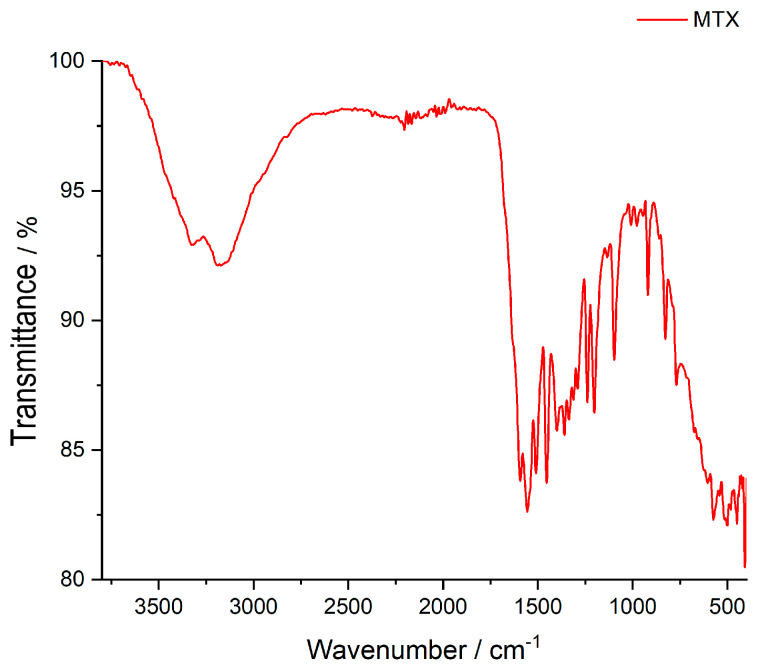
FTIR spectrum of Methotrexate.

**Figure 3 polymers-15-04325-f003:**
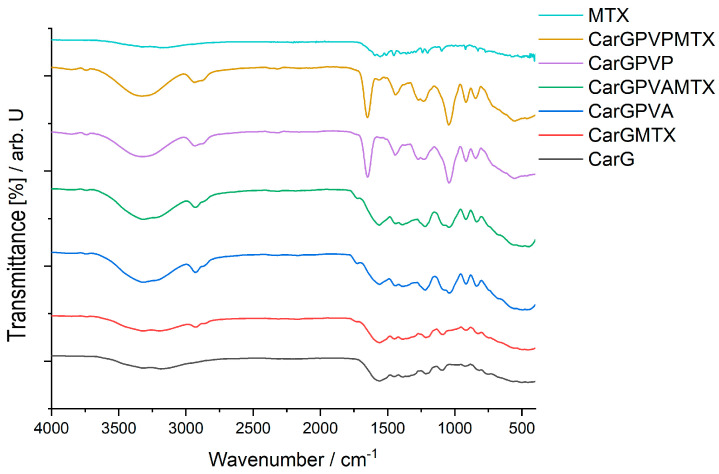
FT-IR spectra of carrageenan-based membranes: **MTX**; **CarGPVPMTX**; **CarGPVP**; **CarGPVAMTX**; **CarGPVA**; **CarGMTX**; **CarG**; **(*Legend***: **Car**–k-carrageenan; **G**–glycerol; **PVA**–polyvinyl alcohol; **PVP**–polyvinylpirrolydone, **MTX**–methotrexate).

**Figure 4 polymers-15-04325-f004:**
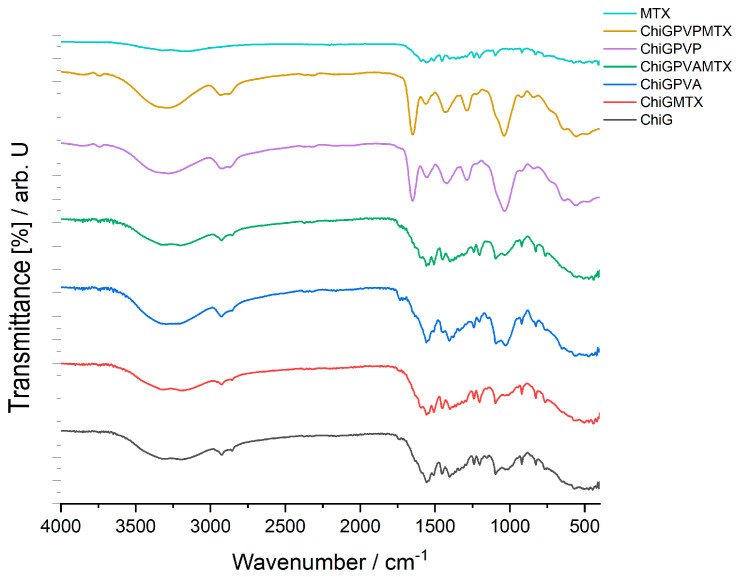
FT-IR spectra of chitosan-based membranes–**MTX**; **ChiGPVPMTX**; **ChiGPVP**; **ChiGPVAMTX**; **ChiGPVA**; **ChiGMTX**; **ChiG (*Legend***: **Chi**–chitosan; **G**–glycerol; **PVA**–polyvinyl alcohol; **PVP**–polyvinylpirrolydone; **MTX**–methotrexate).

**Figure 5 polymers-15-04325-f005:**
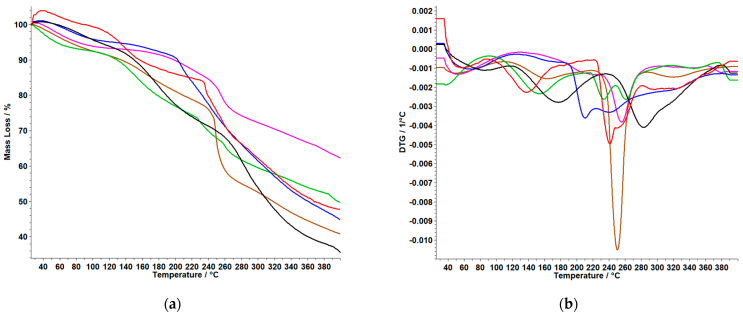
TG (**a**) and DTG (**b**) curves of Car-based membranes: ***CarGPVP(−)**; CarGPVPMTX (−);* CarG (−)**; ***CarGMTX (−)*; CarGPVA *(−)***; ***CarGPVAMTX (−) **(Legend***: **Car**–k-carrageenan; **G**–glycerol; **PVA**–polyvinyl alcohol; **PVP**–polyvinylpirrolydone; **MTX**–methotrexate).

**Figure 6 polymers-15-04325-f006:**
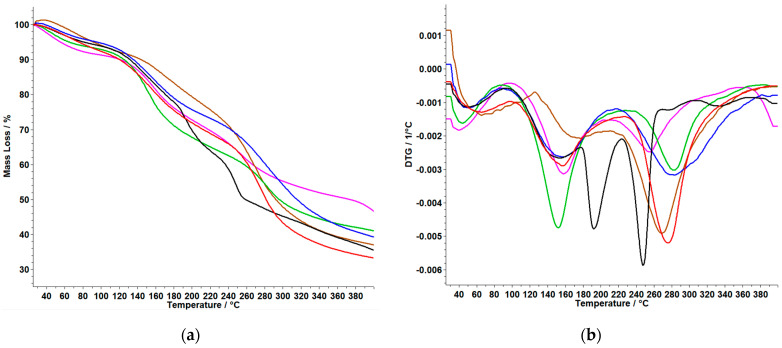
TG (**a**) and DTG (**b**) curves of **Chi***-*based membranes: **ChiGPVP (−); ChiGPVPMTX (−)**; **ChiGMTX (−)**; **ChiG (−)**; **ChiGPVA (−)***;*
**ChiGPVAMTX (−) (*Legend*: Chi**–chitosan; **G**–glycerol; **PVA**–polyvinyl alcohol; **PVP**–polyvinylpirrolydone; **MTX**–methotrexate).

**Figure 7 polymers-15-04325-f007:**
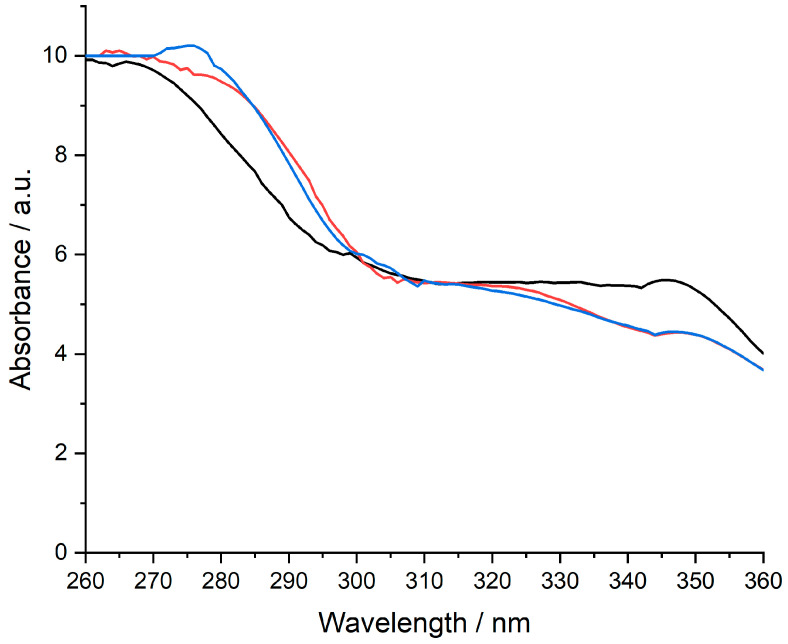
UV–VIS spectrum of **MTX (−)**, **CarGPVAMTX(−),** and **CarGPVPMTX (−) (*Legend*: Car**–k−carrageenan; **G**–glycerol; **PVA**–polyvinyl alcohol; **PVP**–polyvinylpirrolydon**e**; **MTX**–methotrexate).

**Figure 8 polymers-15-04325-f008:**
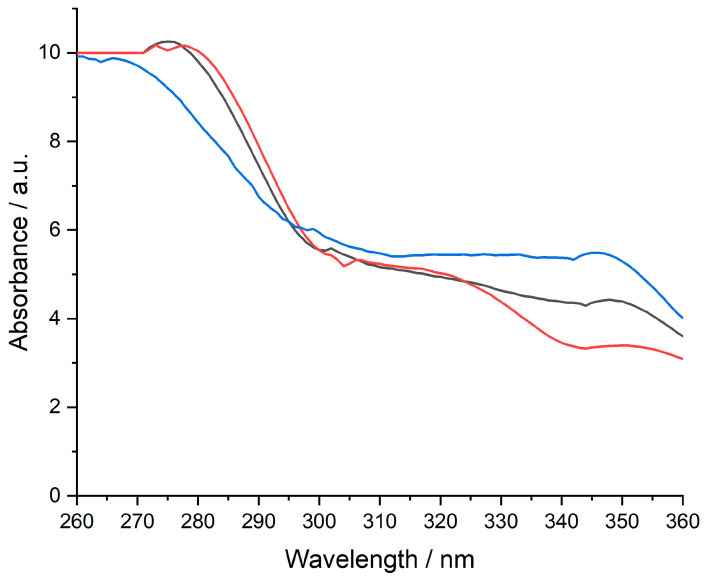
UV–VIS spectrum of **MTX (−)**, **ChiGPVAMTX (−),** and **ChiGPVPMTX (−) (*Legend*: Chi**–chitosan; **G**–glycerol; **PVA**–polyvinyl alcohol; **PVP**–polyvinylpirrolydone; **MTX**–methotrexate).

**Figure 9 polymers-15-04325-f009:**
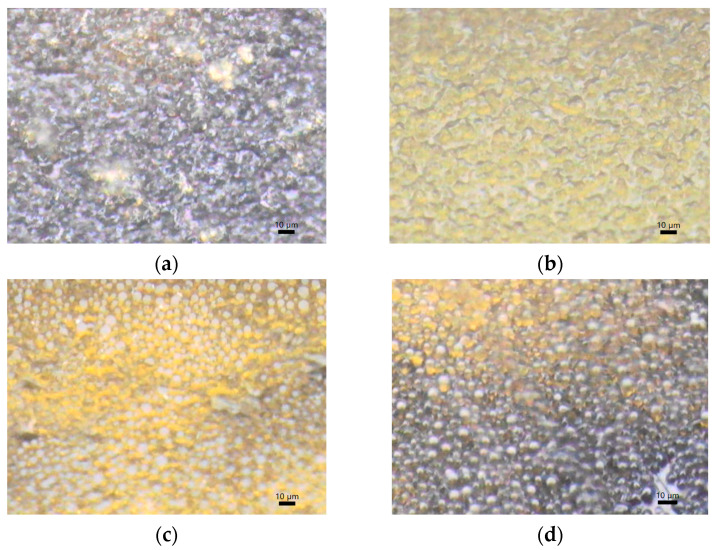
*Appearance of* (**a**) **CarGPVAMTX***;* (**b**) **CarGPVPMTX**; (**c**) **ChiGPVAMTX***;* (**d*)* ChiGPVPMTX**
*(**Legend**:* **Car**–k-carrageenan; **Chi**–chitosan; **G**–glycerol; **PVA**–polyvinyl alcohol; **PVP**–polyvinylpirrolydone; **MTX**–methotrexate).

**Figure 10 polymers-15-04325-f010:**
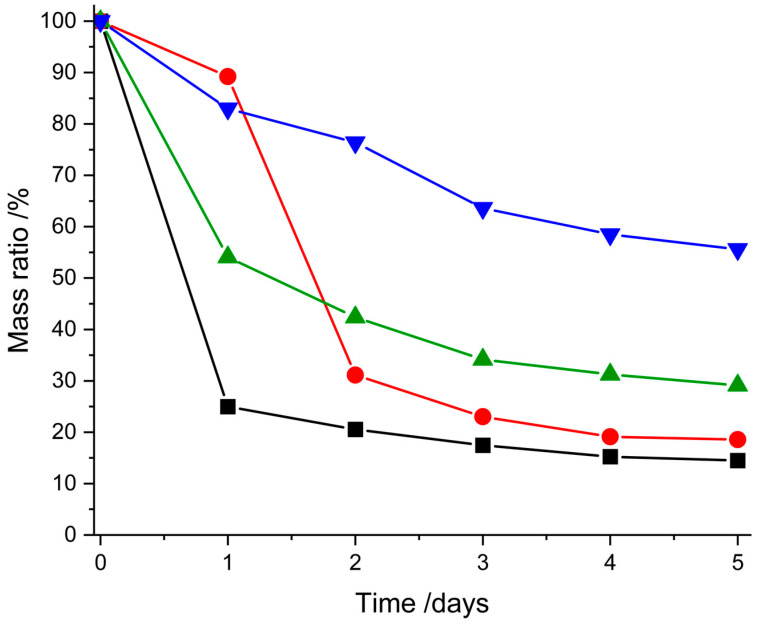
Degradation of membranes in lysozyme at 37 °C (**CarGPVPMTX (−)**, **CarGPVAMTX (−)**, **ChiGPVPMTX (−)**, **ChiGPVAMTX (−)**) (***Legend: Car****–*k-carrageenan*; **Chi**–*chitosan*; **G**–*glycerol*; **PVA**–*polyvinyl alcohol*; **PVP**–*polyvinylpirrolydone; ***MTX****–*methotrexate).

**Table 1 polymers-15-04325-t001:** Membrane synthesis.

	Membrane ID	Car	Chi	G	Span 80	PVA	PVP	MTX
		mg	mg	mL	mL	mg	mg	mg
1	**CarG**	10	/	0.3	/	/	/	/
2	**CarS**	10	/	/	0.3	/	/	/
3	**CarGMTX**	10	/	0.3	/	/	/	1
4	**CarSMTX**	10	/	/	0.3	/	/	1
5	**ChiG**	/	10	0.3	/	/	/	/
6	**ChiS**	/	10	/	0.3	/	/	/
7	**ChiGMTX**	/	10	0.3	/	/	/	1
8	**ChiSMTX**	/	10	/	0.3	/	/	1
9	**CarGPVA**	5	/	0.3	/	5	/	/
10	**CarSPVA**	5	/	/	0.3	5	/	/
11	**CarGPVAMTX**	5	/	0.3	/	5	/	1
12	**CarSPVAMTX**	5	/	/	0.3	5	/	1
13	**ChiGPVA**	/	5	0.3	/	5	/	/
14	**ChiSPVA**	/	5	/	0.3	5	/	/
15	**ChiGPVAMTX**	/	5	0.3	/	5	/	1
16	**ChiSPVAMTX**	/	5	/	0.3	5	/	1
17	**CarGPVP**	5	/	0.3	/	/	5	/
18	**CarSPVP**	5	/	/	0.3	/	5	/
19	**CarGPVPMTX**	5	/	0.3	/	/	5	1
20	**CarSPVPMTX**	5	/	/	0.3	/	5	1
21	**ChiGPVP**	/	5	0.3	/	/	5	/
22	**ChiSPVP**	/	5	/	0.3	/	5	/
23	**ChGPVPMTX**	/	5	0.3	/	/	5	1
24	**ChSPVPMTX**	/	5	/	0.3	/	5	1

***(Legend:* Car**–k-carrageenan; **Chi**–chitosan; **G**–glycerol; **S**–Span 80; **PVA**–polyvinyl alcohol; **PVP**–polyvinylpirrolydone; **MTX**–methotrexate).

**Table 2 polymers-15-04325-t002:** Appearance of membranes.

	Ingredient ID	MembraneAppearance	Aspect		Ingredient ID	MembraneAppearance	Aspect
1	**CarG**	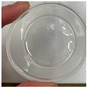	Clear, elastic, homogeneous	5	**ChiG**	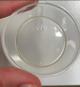	Clear, elastic, homogeneous
2	**CarS**	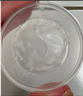	Clear, rigid, inhomogeneous	6	**ChiS**	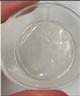	Clear, rigid, homogeneous
3	**CarGMTX**	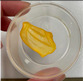	Unclear, elastic, homogeneous	7	**ChiGMTX**	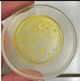	Clear, elastic, homogeneous
4	**CarSMTX**	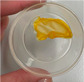	Unclear, rigid, inhomogeneous	8	**ChiSMTX**	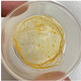	Clear, rigid, inhomogeneous
9	**CarGPVA**	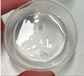	Clear, elastichomogeneous	13	**ChiGPVA**	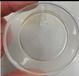	Clear, elastic, homogeneous
10	**CarSPVA**	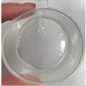	Clear, rigid, homogeneous	14	**ChiSPVA**	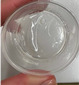	Clear, rigid, homogeneous
11	**CarGPVAMTX**	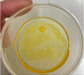	Clear, elastic, homogeneous,	15	**ChiGPVAMTX**	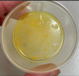	Clear, elastic, homogeneous,
12	**CarSPVAMTX**	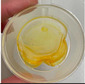	Unclear, rigid, inhomogeneous	16	**ChiSPVAMTX**	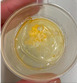	Unclear, rigid, inhomogeneous
17	**CarGPVP**	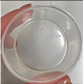	Clear, elastic, homogeneous	21	**ChiGPVP**	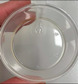	Clear, elastic, homogeneous
18	**CarSPVP**	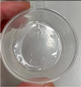	Clear, rigid, homogeneous	22	**ChiSPVP**	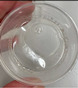	Clear, rigid, homogeneous
19	**CarGPVPMTX**	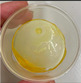	Clear, elastic, homogeneous	23	**ChiGPVPMTX**	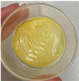	Clear, elastic, homogeneous
20	**CarSPVPMTX**	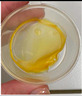	Unclear, rigid, inhomogeneous	24	**ChiSPVPMTX**	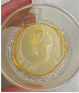	Unclear, rigid, inhomogeneous

***(Legend:* Car**–k-carrageenan; **Chi**–chitosan; **G**–glycerol; **S**–Span 80; **PVA**–polyvinyl alcohol; **PVP**–polyvinylpirrolydone; **MTX**–methotrexate).

**Table 3 polymers-15-04325-t003:** (**a**) Wavenumber values of the most important peaks in FTIR spectra of analyzed carrageenan-based membranes; (**b**) Wavenumber values of the most important peaks in FTIR spectra of analyzed chitosan-based membranes.

	(a)						
Characteristic Bonds and Movement	Compound						
	MTX	CarG	CarGMTX	CarGPVA	CarGPVAMTX	CarGPVP	CarGPVPMTX
		Wavenumber of Most Relevant Peaks (cm^−1^)					
O-H	3351	3316	3340	3300	3300	3320	3320
N-H	3190		3247	3190	3230		
C-H	2950	2940	2952	2926	2927	2929	2932
C=O	1670–1601	1640	1670	1640	1670	1642	1648
N-H	1555–1508	-	1557 1508	1555 1507	1558 1507	1558 1506	1558 1506
CH_3_ and CH_2_ deformation vibration	-	1419 B 1368 B	1400 1358	1401 1375	1401 1376	1423 1374	1423 1374
-C-O-	1453–1201		1453	1452	1450–1205	1459	1459
C-N stretch vibration	1097	1202 A	1202	1204	1095	1035	1112
C-H sym. deformation vibration	-	1160	-	-	1160	1159	1160
C-O stretching, linkage of 3,6-anhydro-D-galactose	-	1032 919	1096 919	1095 919	1035 919	921	920
C-O-SO_3_ of D-galactose-4, sulfate	-	827	827	827	827	843	843
Vibration of -C-C skeleton	-	920	920	920	920	925	924
	**(b)**						
**Characteristic Bonds and Movement**	**Compound**						
	**MTX**	**ChiG**	**ChiGMTX**	**ChiGPVA**	**ChiGPVAMTX**	**ChiGPVP**	**ChiGPVPMTX**
		**Wavenumber of Most Relevant Peaks (cm^−1^)**					
O-H	3351	3300	3300	3270	3300	3296	3300
N-H	3190	3230	3189		3240		
C-H	2950	2925	2926	2926	2925	2923	2927
C=O	1670−1601	1650	1650	1734	1734	1650	1652
N-H	1555–1508	1557	1558	1565 1508	1558 1507	1558	1558
-C-O-	1453–1201	1401	1399	1404	1399	1458	1459
C-N stretching vibration	1097	1323	1320	-	1100	-	1100
Asymmetric bridge oxygen C-O-C stretching	-	1153	-	-	-	1152	1152
Vibration of -C-C skeleton	-	1011 920	920	920	920	925	924

(**a**) ***(Legend*:** A: -S=O stretching vibration; B: bending of O-H group; **Car**–k-carrageenan; **G**–glycerol; **PVA**–polyvinyl alcohol; **PVP**–polyvinylpirrolydone; **MTX**–methotrexate). (**b**) ***(Legend***: **Chi**–chitosan; **G**–glycerol; **PVA**–polyvinyl alcohol; **PVP**–polyvinylpirrolydone; **MTX**–methotrexate).

**Table 4 polymers-15-04325-t004:** (**a**) Thermoanalytical data of membranes based on **Car**; (**b**) Thermoanalytical data of membranes based on **Chi**.

(a)						
Membranes	Process	Temperature (°C)			Mass Loss (%)	Total Mass Loss (%)
		Initial	Final	Max		
**CarG**	I	225	252	237	8.70	57.70
**CarGPVP**	I	118	186	154	12.50	48.41
	II	222	252	230	6.33	
	III	254	277	263	5.15	
**CarGPVA**	I	180	223	192	14.15	63.51
	II	227	263	256	13.00	
**CarGMTX**	I	237	271	247	20.25	58.87
**CarGPVPMTX**	I	240	276	254	9.75	37.12
**CarGPVAMTX**	I	197	233	207	10.7	54.66
(**b**)						
**Membranes**	**Process**	**Temperature (°C)**			**Mass Loss (%)**	**Total Mass Loss (%)**
		**Initial**	**Final**	**Max**		
**ChiG**	I	122	173	157	12.82	66.57
	II	238	320	278	26.16	
**ChiGPVP**	I	106	190	152	23.09	58.39
	II	250	320	286	14.45	
**ChiGPVA**	I	145	208	172	14.80	62.83
	II	254	320	285	19.65	
**ChiGMTX**	I	229	304	270	26.4	63.5
**ChiGPVPMTX**	I	108	200	159	18.04	51.09
	II	223	289	258	15.52	
**ChiGPVAMTX**	I	106	212	160	20.00	60.93
	II	244	320	285	19.65	

(**a**) (***Legend**: **Car**–k-carrageenan; **G**–glycerol; **PVA**–polyvinyl alcohol; **PVP**–polyvinylpirrolydone, **MTX**–methotrexate.*) (**b**) (***Legend**:*
**Chi**–chitosan; **G**–glycerol; **PVA**–polyvinyl alcohol; **PVP**–polyvinylpirrolydone; **MTX**–methotrexate.)

**Table 5 polymers-15-04325-t005:** Swelling ratio of the membranes.

*Membranes*	*SR* (%)
**CarGPVAMTX**	-
**CarGPVPMTX**	-
**ChiGPVAMTX**	111.11
**ChiGPVPMTX**	107.57

(***Legend:* Car***–*k-carrageenan; **Chi***–*chitosan; **G***–*glycerol; **PVA***–*polyvinyl alcohol; **PVP***–*polyvinylpirrolydone; **MTX***–*methotrexate).

**Table 6 polymers-15-04325-t006:** Water vapor transmission rate for the membranes.

*Membranes*	*WVTR* (g·m^−2^·d^−1^)
**CarGPVAMTX**	1389.81
**CarGPVPMTX**	1492.36
**ChiGPVAMTX**	1319.74
**ChiGPVPMTX**	1353.82

(***Legend:* Car***–*k-carrageenan; **Chi***–*chitosan; **G***–*glycerol; **PVA***–*polyvinyl alcohol; **PVP***–*polyvinylpirrolydone; **MTX***–*methotrexate).

## Data Availability

The data presented in this study are available on request from the corresponding author.

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
