# Peer review of "Synthesis and Characterization of Polymer-Based Membranes for Methotrexate Drug Delivery"

_polymers, 2023, doi:10.3390/polym15214325_

Round 1
Reviewer 1 Report
Comments and Suggestions for Authors
I would suggest the following aspects to be considered:
Title: "Preliminary study of polymer-based membrane for Methotrexate drug delivery". Why preliminary? You did not mention that a companion article is coming. I would suggest instead: “Synthesis and characterization of polymer-based membranes for Methotrexate drug delivery"
Abstract : « …Methotrexate or amethopterin or 4-amino-N10-methyl pteroylglutamic acid is used for autoimmune diseases, as well as certain malignancies…”. Is this chemical a medicine or drug? If yes, it must be denoted. I suggest adding “for treating” before autoimmune diseases.
Abstract: “…Drug delivery systems, which are based on biopolymers, can be developed to improve the therapeutic and pharmacological properties of topically administered drugs…”. What kind of improvement, or advantage presents a biopolymer? Readers must understand everything.
Keywords: biopolymers; κ-carrageenan; chitosan; methotrexate; drug delivery; marine polysaccha- 26 rides; FT-IR; thermal analysis; UV-VIS. All key words must be present in the title of the manuscript, thus: κ-carrageenan; chitosan; marine polysaccharides; FT-IR; thermal analysis; UV-VIS. Are not suitable keywords. One way to fix this is to modify the title of the manuscript.
Table 1, Table 2, are not referenced in the manuscript text!
Materials and Methods: Please provide full information on the following analytical instruments. Readers have to be able to reproduce the experiments with no further information: FT-IR; ATR (number of reflections of the internal reflection element); Thermogravimetric analysis (TGA); Differential Scanning Calorimetry (DSC); UV-VIS spectrophotometry.
Experimental section: Please provide full details on the experimental procedures. Readers have to be able to reproduce all experiments and measurements.
Legends to figures and tables: Avoid absolutely using abbreviations. Legends for all figures and tables must present full information about their content so that readers can better understand without referring to the text of the article.
Line 89: « …UV-Vis and microscopy…”. Please rephrase.
Line 143: Write “FTIR-ATR” instead of “FTIR_ATR” (Attenuated Total Reflection)
How was fixed the problem of OH overlapping bands? Water absorbs light significantly producing huge overlapping bands.
Line 213, and line 217: Please, fix the spelling mistakes.
Line 256: “…From the two figures xxx-xxx that showing the DSC curves for the pure compound…”. What is the meaning of xxx-xxx?
Line 258 – 259: “…DSC analysis performed on MTX shows, in nitrogen atmosphere, at a temperature 258 range of 25-400°C, shows four thermal processes (Figure 8 and 9)…”. This phrase is totally not understandable. Is that a “discussion”? Authors have to discuss these results and provide clear conclusions about.
Line 292-295: I suggest rephrasing the text of these lines as they are not clear.
Line 303-304: “…The two membranes validated by this study can be tested later to optimize and validate the transdermal release of the active substance…”. Please mention here these two validated membranes.
Comments on the Quality of English LanguageDear authors,
There are writing inaccuracies in the Summary and Conclusion sections, and also a bit everywhere. I suggest you reorganize the ideas appropriately and especially rewrite the summary which constitutes the main face of your manuscript. When you write a text, think about the readers by asking yourself a question: “Will they understand?” Simply put, you have to get in their face.
Good luck
Author Response
I would suggest the following aspects to be considered:
Title: "Preliminary study of polymer-based membrane for Methotrexate drug delivery". Why preliminary? You did not mention that a companion article is coming. I would suggest instead: “Synthesis and characterization of polymer-based membranes for Methotrexate drug delivery"
R: The title was changed according to reviewer suggestion as follows: ”Synthesis and characterization of polymer-based membranes for Methotrexate drug delivery".
Abstract : « …Methotrexate or amethopterin or 4-amino-N10-methyl pteroylglutamic acid is used for autoimmune diseases, as well as certain malignancies…”. Is this chemical a medicine or drug? If yes, it must be denoted. I suggest adding “for treating” before autoimmune diseases.
R: For treating” was added to the text before” autoimmune diseases”, according to the reviewer's suggestion.
Abstract: “…Drug delivery systems, which are based on biopolymers, can be developed to improve the therapeutic and pharmacological properties of topically administered drugs…”. What kind of improvement, or advantage presents a biopolymer? Readers must understand everything.
R: According to literature, biopolymers have received significant attention in a variety of applications that require biodegradable and sustainable solutions. The development of drug delivery strategies to boost the activity of bioactive substances is still a crucial strategy for achieving disease treatment, and progress in this field has been enormous. Advantages of using biopolymers included: support for the growth of new tissue, inhibition of the activity of cells, guided tissue response, enhancement of the cell-cell interaction and the resulting stimulation of the cells, inhibition of activation or attachment of cells, inhibiting a biological reaction. Biopolymers improve the therapeutic effect of drugs, mainly by improving their biodistribution and modulating drug release. These advantages were included in the text of the manuscript.
Keywords: biopolymers; κ-carrageenan; chitosan; methotrexate; drug delivery; marine polysaccha- 26 rides; FT-IR; thermal analysis; UV-VIS. All key words must be present in the title of the manuscript, thus: κ-carrageenan; chitosan; marine polysaccharides; FT-IR; thermal analysis; UV-VIS. Are not suitable keywords. One way to fix this is to modify the title of the manuscript.
R: The keywords have been changed as suggested by the reviewer as follows: Methotrexate, membranes, drug delivery, polymers, synthesis.
Table 1, Table 2, are not referenced in the manuscript text!
R: Table 1 and Table 2 have been referenced in the manuscript text.
Materials and Methods: Please provide full information on the following analytical instruments. Readers have to be able to reproduce the experiments with no further information: FT-IR; ATR (number of reflections of the internal reflection element); Thermogravimetric analysis (TGA); Differential Scanning Calorimetry (DSC); UV-VIS spectrophotometry.
R: The information of the analytical instruments was completed as follow:
For FTIR ATR: ATR accessory equipped with a single reflection diamond ATR crystal on ZnSe plate was used for all the analysis.
For Thermogravimetric analysis: For calibration on TGA/DSC3+ calibration standards of In (12.29 mg), Al (5.52 mg), Au (41.01) and Pd (26.67 mg) were used
At the suggestion of the reviewer, all information in the manuscript has been completed.
Experimental section: Please provide full details on the experimental procedures. Readers have to be able to reproduce all experiments and measurements.
R: The text of the manuscript contains full details of the experimental procedures.
Legends to figures and tables: Avoid absolutely using abbreviations. Legends for all figures and tables must present full information about their content so that readers can better understand without referring to the text of the article.
R: The legends to the figures and tables were added in the manuscript. The abbreviations in the figures have been deleted. See manuscript with track change.
Line 89: « …UV-Vis and microscopy…”. Please rephrase.
R: The text has been reformulated.
Line 143: Write “FTIR-ATR” instead of “FTIR_ATR” (Attenuated Total Reflection)
R: The text has been corrected.
How was fixed the problem of OH overlapping bands? Water absorbs light significantly producing huge overlapping bands.
R: The text has been corrected.
Line 213, and line 217: Please, fix the spelling mistakes.
R: The spelling mistakes were corrected.
Line 256: “…From the two figures xxx-xxx that showing the DSC curves for the pure compound…”. What is the meaning of xxx-xxx?
R: Since the results of the DSC analysis were not sufficiently clear, we decided to eliminate this method of analysis and to supplement the results with characterization tests.
Line 258 – 259: “…DSC analysis performed on MTX shows, in nitrogen atmosphere, at a temperature 258 range of 25-400°C, shows four thermal processes (Figure 8 and 9)…”. This phrase is totally not understandable. Is that a “discussion”? Authors have to discuss these results and provide clear conclusions about.
R: The results of the DSC analysis were removed.
Line 292-295: I suggest rephrasing the text of these lines as they are not clear.
Line 303-304: “…The two membranes validated by this study can be tested later to optimize and validate the transdermal release of the active substance…”. Please mention here these two validated membranes.
Reviewer 2 Report
Comments and Suggestions for Authors
The authors' work is undoubtedly dedicated to the topical and interesting subject of transdermal drug delivery. At the same time, the study itself evokes contradictory feelings. It seems to have been written in a hurry, without proper proofreading. The English needs to be improved. The main point is the lack of a clearly stated research aim and the tasks required to achieve this aim. I believe that major revisions are required before publication.
General comments:
1. Line 178, "There is no interaction in this type of membrane". It should be made clear whether we are talking about physical or chemical interactions. The former (van der Waals, hydrogen bonding) are clearly present and can also affect the compatibility of components.
2. Lines 200-201: "The FTIR study led us to the idea of possible interactions within the membrane". The discussion of this issue should be as focused as possible, as it is one of the main objectives of the study, based on the title of the paper and the abstract. The functional groups involved in the reactions should be considered and the possible reactions themselves should be shown, discussing how this affects the IR spectrum.
One gets the feeling that the main purpose of the paper was simply to record spectra and thermograms, without proper discussion of them. In all cases, the authors limit themselves to simply stating the values found, without any analysis or connection to the objective.
3. It is recommended to clarify, the purpose of TG and DSC studies in the high temperature region (greater than 100 oC) for transdermal delivery systems?
4. Line 219: readers' attention is drawn to the presence of two "thermal events are accompanied by a loss of mass up to T = 132.5 °C" before melting, the origin of which is not discussed.
5. Line 227, "From the TG and DTG curves for membrane base". Please show the DTG curve in Fig. 6.
6. Figure 10: It would be good to add the spectrum of membranes without MTX. How were the membranes recorded?
7. Line 284-285, "The images obtained with the help of the microscope indicate a homogeneous distribution of both the polymer structure and the active substance in the case of the..." Is the MTX distribution really visible?
8. Conclusions, line 297-300, "...studies by TG/DSC and FTIR methods were performed, which concluded that some membrane bases are unsuitable for MTX incorporation" It is better to indicate specific samples and reasons.
Among the minor comments are:
9. Page 2 line 56: it is common to denote optical isomers by capital letters D and L.
10. No reference to Figures 1 and 4, Tables 1 and 2 in the text of the paper.
11. Page 13 line 256: Indicate figure numbers
12. Page 3 line 101 and further down: the authors talk about copolymers, meaning PVA and PVP. This is a bit misleading; it is better to use the term “polymer”.
13. Table 1: it is better to first introduce the reader to the accepted designations of the samples. Although they are intuitive, there are some disagreements. For example, glycerol in line 109 is labelled as Gly, it is also labelled as "G" in the table.
14. In Table 2, in the “Aspect” column, it is recommended to follow the same way of characterization for all samples
15. Page 5, line 139-140 and Table 2 sample #2 contradict each other.
16. Table 2: "Ingrediant ID" should be "Ingredient ID"
17. p.14 line 277 the word "pick" should be "peak".
18. p.6 line 152 the word "amidic" should be "amide".
19. Table 3a: the C=O band at 1592 cm-1 looks like an error.
20. The sentence on lines 164-165 repeats the sentence on lines 160-161.
Line 185: "is shown in Figure 3" should be "is shown in Figure 4".
Best regards,
Reviewer
Comments on the Quality of English LanguageThe comments are in the text of the review
Author Response
The authors' work is undoubtedly dedicated to the topical and interesting subject of transdermal drug delivery. At the same time, the study itself evokes contradictory feelings. It seems to have been written in a hurry, without proper proofreading. The English needs to be improved. The main point is the lack of a clearly stated research aim and the tasks required to achieve this aim. I believe that major revisions are required before publication.
General comments:
- Line 178, "There is no interaction in this type of membrane". It should be made clear whether we are talking about physical or chemical interactions. The former (van der Waals, hydrogen bonding) are clearly present and can also affect the compatibility of components.
R: From the data presented in the manuscript (Table 3a) it appears that the κ-carrageenan-based membrane with PVA and MTX contains all the characteristic peaks of the active ingredient and the main characteristic peaks of the components. We suggest that the components were integrated into membrane by the physical interaction of hydrogen bonding rather than by a chemical reaction. Considering the presence of all the characteristic peaks, we can conclude that the components are compatible according to the FTIR study.
- Lines 200-201: "The FTIR study led us to the idea of possible interactions within the membrane". The discussion of this issue should be as focused as possible, as it is one of the main objectives of the study, based on the title of the paper and the abstract. The functional groups involved in the reactions should be considered and the possible reactions themselves should be shown, discussing how this affects the IR spectrum.
One gets the feeling that the main purpose of the paper was simply to record spectra and thermograms, without proper discussion of them. In all cases, the authors limit themselves to simply stating the values found, without any analysis or connection to the objective.
R: the error in the text has been corrected so that the compatibility of the compounds used can be established according to FTIR study. The presence of all characteristic peaks related to the active ingredient and the constituents was established. The nature of the interactions was also discussed.
- It is recommended to clarify, the purpose of TG and DSC studies in the high temperature region (greater than 100 oC) for transdermal delivery systems?
R: Thermal analysis techniques, like other physico-chemical methods, are used to characterize materials or substances. It is not used in the study in order to analyze only the thermal stability. The analysis of active substances at temperatures above 100 ℃ is currently used in order to establish the decomposition processes, to validate the different polymorphic forms, etc. In our study, the analysis up to 100 ℃ would not have brought data that we could use in order to validate the compatibility of the different mixtures because they could not highlight the stages of MTX mass loss on which the conclusions of the integrity of MTX in the binary mixtures are based studied.
- Line 219: readers' attention is drawn to the presence of two "thermal events are accompanied by a loss of mass up to T = 132.5 °C" before melting, the origin of which is not discussed.
R: The sentence was reformulated and entire thermal behavior of MTX, the active compound was discussed in our previous work.
- Line 227, "From the TG and DTG curves for membrane base". Please show the DTG curve in Fig. 6.
R: The DTG curves were added in figure.
- Figure 10: It would be good to add the spectrum of membranes without MTX. How were the membranes recorded?
R: Figures contain the spectra of membranes without MTX. Infrared spectra were collected using a Shimadzu IRTracer-ATR in the range of 400–4000 cm−1 at 4 cm−1 optical resolution and 20 scan repetitions per analysis. ATR accessory equipped with a single reflection diamond ATR crystal on ZnSe plate was used for all of the analysis.
- Line 284-285, "The images obtained with the help of the microscope indicate a homogeneous distribution of both the polymer structure and the active substance in the case of the..." Is the MTX distribution really visible?
R: Considering that MTX gives yellow color to membranes, these conclusions were also mentioned after FTIR examination and appearance of membranes.
- Conclusions, line 297-300, "...studies by TG/DSC and FTIR methods were performed, which concluded that some membrane bases are unsuitable for MTX incorporation" It is better to indicate specific samples and reasons.
R: This information was presented in the conclusions section of the study.
Among the minor comments are:
- Page 2 line 56: it is common to denote optical isomers by capital letters D and L.
R: The text has been corrected.
- No reference to Figures 1 and 4, Tables 1 and 2 in the text of the paper.
R: The text has been corrected.
- Page 13 line 256: Indicate figure numbers
R: The text has been corrected and the figure was deleted.
- Page 3 line 101 and further down: the authors talk about copolymers, meaning PVA and PVP. This is a bit misleading; it is better to use the term “polymer”.
R: The text has been corrected.
- Table 1: it is better to first introduce the reader to the accepted designations of the samples. Although they are intuitive, there are some disagreements. For example, glycerol in line 109 is labelled as Gly, it is also labelled as "G" in the table.
R: The text has been corrected.
- In Table 2, in the “Aspect” column, it is recommended to follow the same way of characterization for all samples
R: The information was reviewed.
- Page 5, line 139-140 and Table 2 sample #2 contradict each other.
R: The text has been corrected.
- Table 2: "Ingrediant ID" should be "Ingredient ID"
R: The text has been corrected.
- p.14 line 277 the word "pick" should be "peak".
R: The text has been corrected.
- p.6 line 152 the word "amidic" should be "amide".
R: The text has been corrected.
- Table 3a: the C=O band at 1592 cm-1looks like an error.
R: The Table 3a was corrected.
- The sentence on lines 164-165 repeats the sentence on lines 160-161.
R: The text has been corrected.
Line 185: "is shown in Figure 3" should be "is shown in Figure 4".
R: The text has been corrected.
Reviewer 3 Report
Comments and Suggestions for Authors
Dear Authors
The authors in this study present the synthesis of membranes based on anionic polysaccharides and cationic polysaccharides for transdermal delivery of the active ingredient methotrexate, as well as a compatibility study between methotrexate and each of the components used in the prepared membranes. The obtained membranes based on different marine polysaccharides, namely κ-carrageenan and chitosan, for the release of the active ingredient methotrexate were characterized by techniques such as TG, DSC, FTIR, UV-Vis spectrophotometry, and FTIR microscopy. Following the studies, the membranes suitable for the transdermal release of the active substance were validated.
General comments
The authors dismissed the essential characterization of the transdermal drug delivery systems such as:
1- Water absorption capacity,
2- Mechanical stability,
3- Biodegradation rate,
4- Water vapor and oxygen permeability.
Moreover, the authors did not show the drug-release behavior of the selected developed membranes to prove their idea.
An in vitro study of the dug release should be performed and added to the experimental part in addition to the abovementioned characterizations.
In its current form, I can not recommend the manuscript for publication.
A major revision is required before reconsidering the manuscript for publication.
Comments on the Quality of English LanguageA minor revision is required.
Author Response
The authors in this study present the synthesis of membranes based on anionic polysaccharides and cationic polysaccharides for transdermal delivery of the active ingredient methotrexate, as well as a compatibility study between methotrexate and each of the components used in the prepared membranes. The obtained membranes based on different marine polysaccharides, namely κ-carrageenan and chitosan, for the release of the active ingredient methotrexate were characterized by techniques such as TG, DSC, FTIR, UV-Vis spectrophotometry, and FTIR microscopy. Following the studies, the membranes suitable for the transdermal release of the active substance were validated.
General comments
The authors dismissed the essential characterization of the transdermal drug delivery systems such as:
1- Water absorption capacity,
2- Mechanical stability,
3- Biodegradation rate,
4- Water vapor and oxygen permeability.
Moreover, the authors did not show the drug-release behavior of the selected developed membranes to prove their idea.
An in vitro study of the dug release should be performed and added to the experimental part in addition to the abovementioned characterizations.
In its current form, I can not recommend the manuscript for publication.
A major revision is required before reconsidering the manuscript for publication.
R: In the article, all the recommendations were treated with the utmost seriousness and were processed in the text. Additional tests were also carried out that led to better reasoned results. Thanks to these tests, the list of co-authors was completed. We thank the reviewers for the pertinent recommendations.
R: We added the following tests of the membranes: water absorbtion capacity (Swelling ratio), in vitro biodegradation rate and water vapor and oxygen permeability.
Lists of authors and references lists were updated as a result of additional analyzes added to the study.
Round 2
Reviewer 1 Report
Comments and Suggestions for Authors
I have no more comments
Comments on the Quality of English LanguageNC
Reviewer 2 Report
Comments and Suggestions for Authors
Most of the reviewer's comments have been addressed.
Reviewer 3 Report
Comments and Suggestions for Authors
Dear Authors
The authors respond to almost all the raised comments, and additional experimental work with discussion has been added in addition to supportive references. The revised manuscript's current form is more transparent and convenient for readers.
Accordingly, I can recommend the publication of the revised manuscript.
Greetings
Comments on the Quality of English LanguageA minor revision is required.